# New UB006 Derivatives With Higher Solubility and Cytotoxic Activity in Ovarian Cancer Cells

**DOI:** 10.3390/ph18020194

**Published:** 2025-01-31

**Authors:** Marc Reina, Xavier Ariza, Dolors Serra, Jordi Garcia, Laura Herrero

**Affiliations:** 1Department of Biochemistry and Physiology, School of Pharmacy and Food Sciences, Universitat de Barcelona, E-08028 Barcelona, Spain; 2Department of Inorganic and Organic Chemistry, School of Chemistry, Universitat de Barcelona, E-08028 Barcelona, Spain; 3Institut de Biomedicina de la Universitat de Barcelona (IBUB), Universitat de Barcelona, E-08028 Barcelona, Spain; 4Centro de Investigación Biomédica en Red (CIBER) de Fisiopatología de la Obesidad y Nutrición (CIBEROBN), Instituto de Salud Carlos III, E-28029 Madrid, Spain

**Keywords:** UB006, fatty acid synthase, cytotoxicity, cancer

## Abstract

Background/Objectives: The compound (±)-**UB006** ((4SR,5SR)-4-(hydroxymethyl)-3-methylene-5-octyldihydrofuran-2(3H)-one) is a promising anti-cancer molecule. The enantiomer (–)-**UB006** displays a potent cytotoxic effect in several tumor cell lines, particularly the ovarian cancer OVCAR-3 cell line, with a 40-fold increase in potency compared with the fatty acid synthase (FAS) inhibitor C75. Furthermore, in vivo, (–)-**UB006** reduced the tumor burden in neuroblastoma xenografts. This effect was attributed to FAS inhibition and upregulation of apoptotic markers. However, CoA adducts of UB006 presented low solubility. Methods: We synthesized several (±)-**UB006** derivatives by elongating the carbon chain of the primary alcohol and/or by adding hydroxyl groups with the aim of finding more potent and soluble anti-cancer compounds. Results: Our results showed a decrease in cytotoxicity when the carbon chain was elongated by more than two carbons. However, ethyl or propyl polyhydroxylated four-branched compounds showed an increased or maintained potency and solubility. The most promising compound was (±)-**UB035** (IC50: 2.1 ± 0.2 µM), with a 2.5-fold increase in cytotoxicity in the OVCAR-3 cell line and a >4-fold increase in solubility (>2 mM) compared with (±)-**UB006**.

## 1. Introduction

Fatty acid synthase (FAS) is a pivotal enzyme in de novo fatty acid synthesis that plays a key role in fueling the enhanced lipid requirements of rapidly proliferating cancer cells. Driven by these observations, FAS inhibition constitutes an attractive strategy to disrupt these vital processes and selectively impede tumor progression [1,2]. Among different alternatives, **C75** and **UB006** share comparable biological targets that underscore their potential as FAS inhibitors and inducers of apoptosis in cancer cells [2,3,4]. Specifically, **C75** binds to the active site of FAS, interfering with its catalytic activity and inhibiting the conversion of acetyl-CoA and malonyl-CoA to long-chain fatty acids [5], thus reducing mitochondrial respiration and ATP production. This interference not only disrupts the cellular production of lipids necessary for membrane integrity and signaling but also prevents the activation of downstream signaling cascades, such as AMP-activated protein kinase [6], and other apoptotic regulators, including members of the Bcl-2 protein family [7] and caspases [8], and inhibits PI3K-mTORC1 [9]. In addition, neuroblastoma xenografts treated with **UB006** showed a reduction in cell proliferation and an increase in neural differentiation [4]. This dual impact on lipid homeostasis and cellular energy dynamics aligns **C75** and **UB006** as compounds with overlapping molecular targets, suggesting a commonality in their mechanisms of action in the pursuit of selectively targeting cancer cells and hindering their proliferative potential.

The compound **C75** also acts centrally in the hypothalamus, specifically on neurons that regulate feeding behavior, altering their sensitivity to peripheral signals, such as the fasting-induced hormone ghrelin [10]. This alteration in neural signaling results in reduced food intake and increased energy expenditure. However, we previously showed that this effect was only observed with the (+)-**C75** enantiomer due to its inhibitory effect on carnitine palmitoyltransferase 1, a key enzyme involved in mitochondrial fatty oxidation [5].

Solubility is a key physicochemical property for the success of any drug candidate, affecting significantly both in vivo and in vitro assays. The compound **C75** is considered insoluble in water and presents low solubility in basic aqueous media (<0.1 mg/mL) [11] due to its lipophilic nature. Unfortunately, the requirement for co-solvents may compromise cell viability in vitro and limit drug success in vivo. Among different small molecule alternatives, **C75** and **UB006** show promising anti-cancer properties [2,3,4]. However, we have previously reported that both **C75** and **UB006** require a solution containing 10% DMSO plus 20% Kolliphor^®^ [12], thus limiting their use in aqueous environments. Therefore, there is a need to develop anti-cancer analogs with a higher solubility. The present study describes a novel approach in the synthesis of **UB006** analogues. Our focus centered on the rational design and synthesis of **UB006** derivatives with the overarching goal of improving their therapeutic potential. Analogues were systematically engineered to enhance cytotoxicity and FAS inhibition and optimize the pharmacokinetic properties. We primarily developed stereoisomeric mixtures of the analogues, as the materials required for their synthesis were more readily available. The most promising compound was (±)-**UB035**, which showed enhanced cytotoxic effects against the cancer cell line OVCAR-3 and was more soluble than (±)-**UB006**.

## 2. Results and Discussion

### 2.1. Synthesis of ***UB006*** and Its Novel Derivatives

A series of UB006 analogues (Figure 1, Table 1) with a modified hydroxymethyl chain was synthesized, and their activities were tested.

Synthesis of (±)-**UB006**

An improved method for (±)-**UB006** synthesis was achieved. First, an efficient acyl radical addition of nonanal to dimethyl maleate using Co(II) as a catalyst produced adduct (±)-**2** with a yield of 96% [13]. The ketone group was selectively reduced to alcohol with NaBH_4_ and then lactonized in situ to a mixture of cis and trans isomeric lactones under an acidic medium. This mixture was isomerized to produce the more stable trans isomer through treatment with base (1,8-diazabicyclo(5.4.0)undec-7-ene; DBU) under anhydrous conditions to produce ꝩ-butyrolactone (±)-trans-**3** as a racemic mixture. The methyl ester was then hydrolyzed in basic media, and the carboxylic compound (±)-trans-**4** was selectively reduced to the corresponding alcohol (±)-trans-**7** using borane-mediated reduction, leaving the lactone functionality intact (Figure 1). A key step in the sequence was the α-methylenation of the resulting lactone, which was successfully achieved through exposure to MMC (Greene’s procedure) [14] to yield (±)-**UB006**.

Aldehyde (±)-trans-**8** was used as the starting material for the synthesis of the new compounds **UB032**, **UB033**, **UB034**, **UB035**, and **UB036** (Figure 2). This compound was easily obtained through Swern oxidation of primary alcohol (±)-trans-**7**.

Synthesis of **UB031**

After a failed attempt to obtain alcohol (±)-trans-**6** through controlled addition of MeMgBr to (±)-trans-**8** at a low temperature, we finally obtained (±)-trans-**6** in a three-step sequence. First, we transformed compound (±)-trans-**4** into its corresponding Weinreb amide (±)-trans-**5**. Then, careful addition of MeMgBr at −20 °C provided the expected methyl ketone in crude form, which was selectively reduced in situ to the desired alcohol (±)-trans-**6** as a mixture of stereoisomers via NaBH_4_ treatment (Figure 1). Synthesis of **UB031** was finally achieved using two-step α-methylenation of the resulting lactone, according to Greene’s procedure. Figure 3 shows the two strategies used for methylenation (MMC or CO_2_/LDA) and the newly synthesized α-methylenebutyrolactones.

Synthesis of **UB032**

N-methylephedrine-catalyzed addition of ZnEt_2_ to aldehyde (±)-trans-**8** according to the classical Soai procedure produced the expected alcohol (±)-trans-**9** but with a moderate yield [15]. An alternative Grignard reaction using ethylmagnesium bromide was unable to yield the desired outcome and was much less selective. Finally, the two-step α-methylenation of the resulting lactone, first introducing an α-carboxylic acid and then the α-methylene group, produced the desired product, **UB032**.

Synthesis of **UB033**

The addition of 1-hexynyl lithium to aldehyde (±)-trans-**8** in THF at a low temperature selectively yielded the desired alcohol (±)-trans-**10**. The resulting product was then easily hydrogenated until saturated to obtain (±)-trans-**11**, and α-methylenation of the resulting lactone using Greene’s procedure yielded compound **UB033**.

Synthesis of **UB034**

The synthesis of **UB034** was initiated through the Grignard addition of vinyl magnesium bromide to aldehyde (±)-trans-**8** at −20 °C to produce allylic alcohol (±)-trans-**12** in moderate yield. The improved performance of this addition compared with previous experiments using butyl or hexyl magnesium bromides is likely attributable to the milder character of vinyl metal reagents. Finally, α-methylenation using Greene’s procedure of the resulting lactone generated a sample of the compound **UB034**.

Synthesis of **UB035**

Treatment of allylic alcohol (±)-trans-**12** with ozone followed by reduction with NaBH_4_ facilitated the isolation of diol (±)-trans-**13** as a mixture of stereoisomers. Further α-methylenation of the resulting compound using Greene’s procedure produced compound **UB035**.

Synthesis of **UB036**

Triol (±)-trans-**14** was easily obtained following a well-known procedure using Os(VIII) oxide dihydroxylation of (±)-trans-**12** as a mixture of stereoisomers. Subsequent α-methylenation of the resulting lactone using Greene’s procedure produced **UB036**.

### 2.2. Cytotoxic Effects on the OVCAR-3 Cell Line

Fatty acids are an important source of energy in cells. Thus, FAS is a relevant target enzyme in rapidly proliferating tissues, such as in cancer [16,17]. To test the anti-tumor activity of the newly synthesized (±)-**UB006** derivatives, cytotoxicity assays were performed in the commonly used ovarian cancer cell line, OVCAR3 (Figure 2). An MTT cytotoxic assay showed that (±)-**UB031** and (±)-**UB035** were better cytotoxic agents than (±)-**UB006**, while (±)-**UB036** maintained a comparable effect. (±)-**UB031** and (±)-**UB035** showed 1.4-fold (IC50 = 3.6 ± 0.2 µM) and 2.5-fold (IC50 = 2.1 ± 0.2 µM) increases in cytotoxicity, respectively, compared with (±)-**UB006** (IC50 = 5.0 ± 0.6 µM). We previously showed that racemic and (–) enantiomers of **UB006** are better cytotoxic agents than racemic **C75** [18]. Differences in the cytotoxic capacity of (±)-**UB006** were observed between the present results and our previous findings [18]. This discrepancy could be due to differences in the passage of the OVCAR3 cell line used, which may have caused partial dedifferentiation, subsequently affecting the cells’ susceptibility to the compounds.

### 2.3. Effect of (±)-***UB006*** and Its Derivatives on FAS Activity

The inhibitory effect of (±)-**C75** on FAS has been previously described by our group (IC50 = 460 ± 56 µM) [5]. Here, we analyzed the effect of racemic mixtures of (±)-**UB006**, and its derivatives, (±)-**UB035** and (±)-**UB036,** on FAS activity. All exhibited comparable FAS inhibitory activity (Figure 3), with IC50 values of 458 ± 49 µM, 565 ± 62 µM, and 458 ± 54 µM, respectively. This indicates that the enhanced cytotoxic activity observed for (±)-**UB035** compared with (±)-**UB006** (Figure 2) was not attributable to differences in FAS inhibition (Figure 3).

### 2.4. Thermodynamic Solubility Assessment of (±)-***UB006*** and Its Derivatives

The solubility of (±)-**UB006** and its derivatives was assessed through turbidimetry using water and 1% DMSO as the solvent and after a 2 h incubation at 37 °C with each compound (Table 2). Turbidimetric analysis showed a biphasic curve, presenting an inflection point at which the compounds failed to solubilize. This effectively created two regression lines, with the intersection corresponding to the solubility of the compound.

The thermodynamic solubility of (±)-**UB006** was estimated to be 550 µM. Elongation of the carbon chain led to progressive loss of solubility, with solubility values of 473 µM, 390 µM, and 76 µM for (±)-**UB031**, (±)-**UB032**, and (±)-**UB033**, respectively. The compound (±)-**UB034** showed a dramatic decrease in solubility (88 µM) compared with (±)-**UB032**, which was likely attributable to the loss of free rotation of the double bond. Derivatives (±)-**UB035** and (±)-**UB036** were soluble at all tested concentrations (25–2000 µM), presenting at least a 4-fold increase in solubility compared with (±)-**UB006**. Importantly, although the increase in solubility of both (±)-**UB035** and (±)-**UB036** did not change FAS activity compared to (±)-**UB006** (Figure 3), it might have contributed to their enhanced cytotoxicity in ovarian cancer cells (Figure 2).

## 3. Materials and Methods

### 3.1. Synthesis of (±)-***UB006*** Derivatives

Dimethyl 2-nonanoylsuccinate, (±)-**2**

CoCl_2_ (0.6 g, 4.6 mmol) was added to a mixture of dimethyl maleate (5 mL, 40 mmol) and n-nonanal (50 mL, 290 mmol). The resulting mixture was stirred for 16 h at 25 °C under a dry O_2_ flow. The reaction mixture was treated with ethyl acetate (AcOEt; 200 mL) and washed with aqueous saturated NaHCO_3_ (2 × 200 mL). The organic layer was dried (MgSO_4_), filtered, and concentrated under reduced pressure. The crude extract was purified through flash chromatography (hexane/CH_2_Cl_2_, 1:1) to yield the desired product (21.94 g, 22.85 mmol, 96%). Oil; Rf = 0.63 (hexane: CH_2_Cl_2_ 2:8); ^1^H NMR (CDCl_3_, 400 MHz): δ 3.99 (dd, J = 6.3, 8.2, 1H), 3.74 (s, 3H), 3.68 (s, 3H), 2.98 (dd, J = 8.2, 17.6, 1H), 2.84 (dd, J = 6.3, 17.6, 1H), 2.75–2.57 (m, 2H), 1.64–1.27 (m, 12H), 0.88 (t, J = 6.8, 3H); ^13^C NMR (CDCl_3_, 101 MHz): δ 204.1, 172.0, 169.2, 54.0, 52.9, 52.2, 43.0, 32.4, 32.0, 29.5, 29.3, 29.2, 23.6, 22.8, 14.3; IR (film): 2954, 2927, 2856, 1741, 1720, 1457, 1437, 1410, 1367, 1262, 1634, 1012, 668.

(2SR,3RS)-Methyl 2-octyl-5-oxotetrahydrofuran-3-carboxylate, (±)-*trans*-3 (HRMS (ESI+) calculated for C_14_H_28_O_4_ [M + H]+ = 257.1747, found = 257.1746) and (2SR,3RS)-2-octyl-5-oxotetrahydrofuran-3-carboxylic acid, (±)-*trans*-4 (HRMS (ESI+) calculated for C_13_H_22_NaO_4_ [M + Na]+ = 265.1410, found = 265.1410) were synthesized following the procedures previously described [18].

(2RS,3RS)-N-methoxy-N-methyl-2-octyl-5-oxotetrahydrofuran-3-carboxamide, (±)-*trans-***5**

A solution of (±)-*trans*-**4** (3.65 g, 15.12 mmol), EDC (4.23 g, 27.22 mmol), and DMAP (1.85 g, 15.12 mmol) was stirred in CH_2_Cl_2_ (35 mL) at 0 °C for 30 min. Then, a solution of N,O-dimethylhydroxylamine (2.95 g, 30.24 mmol) in CH_2_Cl_2_ (10 mL) was added. The mixture was stirred overnight at room temperature. Then, the mixture was quenched with brine (20 mL) and extracted with AcOEt (3 × 30 mL). Afterwards, combined organic phases were dried (MgSO_4_) and concentrated under reduced pressure. Purification of the residue through flash chromatography (CH_2_Cl_2_) yielded the desired product (3.3 g, 11.56 mmol, 75%). Oil; Rf = 0.61 (CH_2_Cl_2_:AcOEt 9:1); ^1^H NMR (400 MHz, CDCl_3_): δ 4.66 (m, 1H), 3.70 (s, 3H), 3.39 (m, 1H), 3.22 (s, 3H), 2.86 (dd, J = 9.4, 17.5, 1H), 2.73 (dd, J = 9.4, 17.5, 1H), 1.78 (m, 2H), 1.56–1.28 (m, 12H), 0.88 (t, J = 6.7, 3H); δ ^13^C NMR (101 MHz, CDCl_3_): δ 175.0, 171.5, 82.4, 61.6, 42.8, 35.2, 33.0, 32.4, 31.8, 29.4, 29.3, 29.2, 25.3, 22.7, 14.1; IR (film): 2924, 2854, 1774, 1659, 1463, 1450, 1177, 1006. MS (ESI+) calculated for C_15_H_27_NO_4_ = 286.2 [M + H]+, found = 286.2.

(2SR,3RS)-4-((RS)-1-hydroxyethyl)-5-octyldihydrofuran-2(3H)-one, (±)-*trans*-**6**

A solution of methylmagnesium bromide in diethyl ether (Et_2_O; 395 µL, 1.19 mmol) was slowly added to a solution of (±)-*trans*-**5** (308 mg, 1.08 mmol) in anhydrous THF (6 mL) at −20 °C under N_2_. The mixture was stirred for 30 min at −20 °C and then allowed to warm up to room temperature overnight. The reaction mixture was quenched with aqueous saturated NH_4_Cl (4 mL) and extracted with AcOEt (3 × 15 mL). Afterwards, combined organic phases were dried (MgSO_4_), filtered, and concentrated under reduced pressure. NaBH_4_ (21 mg, 0.55 mmol) was added to a stirred solution of the crude extract in MeOH (3 mL) at 0 °C, and the mixture was warmed to room temperature. After 1 h, the mixture was quenched with phosphate buffer pH 7 (5 mL) and extracted with AcOEt (3 × 10 mL). Afterwards, combined organic phases were dried (MgSO_4_) and concentrated under reduced pressure. Purification of the residue through flash chromatography (CH_2_Cl_2_/AcOEt 6:4) yielded the desired product, which was a mixture of stereoisomers (0.115 g, 0.47 mmol, 44%). Oil; Rf = 0.30 (CH_2_Cl_2_/AcOEt 6:4); ^1^H NMR (400 MHz, CDCl_3_): δ 4.53–4.37 (m, 1H), 3.93–3.73 (m, 1H), 2.70–2.58 (dd, J = 18.0, 7.7, 1H), 2.58–2.29 (dd, J = 18.0, 9.6, 1H), 2.22–2.12 (m, 1H), 1.82–1.16 (m, 14H), 1.21 (s, 3H), 0.88 (t, J = 6.7, 3H). ^13^C NMR (101 MHz, CDCl_3_): δ 177.2, 176.9, 83.2, 82.7, 69.2, 66.5, 47.4, 47.0, 36.2, 35.2, 31.9, 31.8, 31.6, 29.4, 29.3, 29.2, 25.5, 22.6, 21.8, 14.1. MS (ESI+) calculated for C_14_H_27_O_3_ = 243.2 [M + H]+, found = 243.2.

(4SR,5SR)-4-(Hydroxymethyl)-5-octyldihydrofuran-2(3H)-one, (±)-*trans*-**7** (HRMS (ESI+) calculated for C_13_H_25_O_3_ [M + H]+ = 229.1798, found = 229.1797) and (2SR,3RS)-2-octyl-5-oxotetrahydrofuran-3-carbaldehyde, (±)-*trans*-**8** (HRMS (ESI+) calculated for C_13_H_23_O_3_ [M + H]+ = 227.1638, found = 227.1642) were synthesized following the procedures previously described [18].

(2SR,3RS)-4-((RS)-1-hydroxypropyl)-5-octyldihydrofuran-2(3H)-one, (±)-*trans*-**9**

Diethylzinc (1 M) in heptane (6.6 mL, 6.6 mmol) was slowly added to a solution of (±)-*trans*-**8** (217 mg, 0.95 mmol) and (+)-N-methylephedrine (10 mg, 6% mol) in anhydrous hexane (3 mL) at 0 °C under N_2_. The mixture was stirred for 16 h, warming up to room temperature. The reaction mixture was quenched with aqueous HCl 1N (10 mL) and extracted with CH_2_Cl_2_ (3 × 15 mL). Afterwards, combined organic phases were dried (MgSO_4_), filtered, and concentrated under reduced pressure. Purification of the residue through flash chromatography (hexane/AcOEt, 7:3) produced the desired product as a mixture of stereoisomers (54 mg, 0.21 mmol, 22%). Oil; Rf = 0.42 (hexane/AcOEt 7:3); ^1^H NMR (400 MHz, CDCl_3_): δ 4.41–4.33 (m, 1H), 3.69–3.61 (m, 1H), 2.70–2.60 (dd, J = 18.0, 7.7, 1H), 2.52–2.40 (dd, 1H), 2.25–2.14 (m, 1H), 1.82–1.16 (m, 16H), 1.03 (t, J = 7.4, 3H), 0.88 (t, J = 6.7, 3H). ^13^C NMR (101 MHz, CDCl_3_): δ 177.1, 83.9, 70.2, 45.8, 35.8, 31.8, 29.5, 29.4, 29.2, 29.0, 28.0, 25.8, 22.7, 14.1, 9.8. MS (ESI+) calculated for C_14_H_27_O_3_ [M + H]+ = 242.2, found = 242.1.

(2SR,3RS)-4-((RS)-1-hydroxyhept-2-yn-1-yl)-5-octyldihydrofuran-2(3H)-one, (±)-*trans*-**10**

BuLi (1.6 M) in hexane (0.65 mL, 1.05 mmol) was slowly added to a solution of 1-hexyne (133 µL, 1.16 mmol) in anhydrous THF (3 mL) under N_2_. The mixture was stirred for 30 min at −78 °C. A solution of (±)-*trans*-8, 240 mg, 1.05 mmol) in THF was added at −78 °C, and the mixture was then stirred for 2 h at 0 °C. The reaction was quenched with saturated aqueous NH_4_Cl (4 mL) and extracted with hexane/AcOEt 9:1 (3 × 10 mL). Afterwards, the combined organic phases were dried (MgSO_4_), filtered, and concentrated under reduced pressure. Purification of the residue through flash chromatography (hexane/AcOEt 8:2) yielded the desired product as a mixture of stereoisomers (0.14 g, 0.45 mmol, 43%). Oil; Rf = 0.55 (hexane/AcOEt 8:2); ^1^H NMR (400 MHz, CDCl_3_): δ 4.55–4.44 (m, 1H), 4.42–4.38 (m, 1H), 2.67–2.61 (m, 1H), 2.60–2.5. (dd, J = 18.0, 7.7, 1H), 2.46–2.40 (dd, J = 18.0, 9.6, 1H), 2.23–2.18 (m, 2H), 1.80–1.20 (m, 19H), 0.94–0.88 (m, 6H). ^13^C NMR (101 MHz, CDCl_3_): δ 176.3, 88.3, 88.0, 82.2, 82.0, 78.0, 63.5, 63.1, 46.2, 35.7, 31.8, 30.9, 30.4, 29.7, 29.4, 29.3, 25.3, 22.6, 21.9, 18.2, 14.1, 13.5. MS (ESI+) calculated for C_19_H_33_O_3_ [M + H]+ = 309.2, found = 309.2.

(2SR,3RS)-4-(1-hydroxyheptyl)-5-octyldihydrofuran-2(3H)-one, (±)-*trans*-**11**

A solution of (±)-*trans*-**10** (91 mg, 0.30 mmol) in AcOEt (5 mL) was treated with 5% Pd/C under H_2_, and the suspension was shaken for 5 h at room temperature. The mixture was filtered through Celite^®^, and the solid was washed with AcOEt. Afterwards, combined organic phases were dried (MgSO_4_) and concentrated under reduced pressure to produce the desired product as a mixture of stereoisomers (92 mg, 0.29 mmol, 99%). Oil; Rf = 0.23 (CH_2_Cl_2_); ^1^H NMR (400 MHz, CDCl_3_): δ 4.54–4.39 (m, 1H), 3.68–3.54 (m, 1H), 2.71–2.59 (m, 1H), 2.51–2.41 (dd, J = 18.0, 7.7, 1H), 2.39–2.30 (dd, J = 18.0, 9.6, 1H), 2.23–2.14 (m, 1H), 1.80–1.20 (m, 23H), 0.94–0.88 (m, 6H). ^13^C NMR (101 MHz, CDCl_3_): δ 177.2, 177.0, 83.0, 82.8, 73.3, 70.1, 46.3, 45.6, 36.3, 35.8, 35.39, 35.14, 31.8, 31.7, 29.7, 29.6, 29.48, 29.45, 29.42, 29.37, 29.23, 29.18, 29.15, 28.9, 25.8, 25.58, 25.52, 25.50, 22.6,22.5, 14.1, 14.0. MS (ESI+) calculated for C_19_H_37_O_3_ [M + H]+ = 313.3, found = 313.2.

(2SR,3RS)-4-((RS)-1-hydroxyallyl)-5-octyldihydrofuran-2(3H)-one, (±)-*trans*-**12**

Vinylmagnesium bromide (1 M) THF solution (5.2 mL, 5.2 mmol) was slowly added to a solution of (±)-*trans*-**8** (1.18 g, 4.38 mmol) in anhydrous THF in N_2_ atm. (15 mL). The mixture was stirred at −78 °C for 1 h and allowed to warm to room temperature overnight. The reaction mixture was quenched with aqueous 1 N HCl (10 mL) and extracted with hexane/AcOEt 9:1 (3 × 20 mL). Afterwards, combined organic phases were dried (MgSO_4_), filtered, and concentrated under reduced pressure. Purification of the residue through flash chromatography (hexane/AcOEt 8:2) yielded the desired product as a mixture of stereoisomers (0.223 g, 0.87 mmol, 20%). Oil; Rf = 0.45 (hexane/AcOEt 7:3); ^1^H NMR (400 MHz, CDCl_3_): δ 5.87–5.75 (ddd, J = 10.4, 6.5, 1.4 Hz, 1H), 5.38–5.29 (dd, J = 3.3, 1.2 Hz, 1H), 5.29–5.24 (dt, J = 3.29, 1.17, 1H), 4.19–4.07 (m, 1H), 3.93–3.73 (m, 1H), 2.67–2.33 (dd, J = 18.0, 7.7, 1H), 2.59–2.53 (dd, J = 18.0, 9.6, 1H), 2.33–2.24 (m, 1H), 1.82–1.16 (m, 14H), 0.88 (t, J = 6.7, 3H). ^13^C NMR (101 MHz, CDCl_3_): δ 176.7, 176.6, 138.0, 117.6, 117.4, 82.5, 82.1, 74.4, 72.6, 69.2, 66.4, 45.4, 44.9, 36.0, 35.3, 31.8, 31.3, 29.6, 29.4, 29.3, 29.2, 25.4, 22.6, 14.1. MS (ESI+) calculated for C_15_H_27_O_3_ [M + H]+ = 255.2, found = 255.2.

(2SR,3RS)-4-(1,2-dihydroxyethyl)-5-octyldihydrofuran-2(3H)-one, (±)-*trans*-**13**

A stream of ozone gas was passed through a solution of (±)-*trans*-**11** (456 mg, 1.79 mmol) in CH_2_Cl_2_ (40 mL) at −78 °C until blue coloration persisted. Then, a stream of N_2_ was passed through the mixture for 2 min, and NaBH_4_ (339 mg, 8.96 mmol) and MeOH (20 mL) were added to the mixture and stirred for 1 h at room temperature. The mixture was quenched with saturated aqueous NH_4_Cl (20 mL) and brine (20 mL). The combined aqueous phases were extracted with EtOAc (3 × 50 mL). Afterwards, combined organic phases were dried (MgSO_4_), filtered, and concentrated under reduced pressure. Purification of the residue through flash chromatography (CH_2_Cl_2_/MeOH 98:2) gave the desired diol as a mixture of stereoisomers (187 mg, 0.72 mmol, 40%). Oil; Rf = 0.29 (CH_2_Cl_2_/MeOH 95:5); ^1^H NMR (400 MHz, CD_3_OD): δ 4.47–4.31 (m, 1H), 3.58–3.45 (m, 1H), 3.42–3.31 (m, 2H), 2.62–2.53 (dd, J = 18.0, 7.7, 1H), 2.43–2.33 (dd, J = 18.0, 9.6, 1H), 2.33–2.24 (m, 1H), 1.82–1.16 (m, 14H), 0.88 (t, J = 6.7, 3H). ^13^C NMR (101 MHz, CDCl_3_): δ 178.2, 178.0, 83.4, 83.2, 73.2, 69.8, 64.1, 64.0, 36.2, 35.7, 34.5, 31.6, 29.1, 29.0, 28.9, 25.3, 22.3, 13.0. MS (ESI+) calculated for C_14_H_27_O_4_ [M + H]+ = 259.2, found = 259.3.

(2SR,3RS)-5-octyl-4-(1,2,3-trihydroxypropyl)dihydrofuran-2(3H)-one, (±)-*trans*-**14**

A solution of (±)-*trans*-**12** (100 mg, 0.39 mmol) in THF (4 mL) was added to a solution of K_2_OsO_4_ (5% mol) and 4-methylmorpholine N-oxide (105 mg, 0.78 mmol) in H_2_O/acetone (1:1, 8 mL). The mixture was stirred at room temperature overnight. Then, the mixture was filtered, and HCl 0.5N (10 mL) was added and extracted with AcOEt (3 × 15 mL). The combined organic phases were washed with HCl 1N (40 mL) and dried (MgSO_4_), filtered, and concentrated under reduced pressure. Flash chromatography (CH_2_Cl_2_/MeOH 94:6) of the residue allowed us to obtain the desired triol as a mixture of stereoisomers (0.122 g, 0.42 mmol, 66%). Oil; Rf = 0.26 (CH_2_Cl_2_/MeOH 98:2); ^1^H NMR (400 MHz, CD_3_OD): δ 4.64–4.38 (m, 1H), 3.80–3.70 (m, 1H), 3.60–3.50 (m, 2H), 3.48–3.31 (m, 1H), 2.80–2.42 (m, 3H), 1.86–1.20 (m, 14H), 0.88 (t, J = 6.7, 3H). ^13^C NMR (101 MHz, CDCl_3_): δ 182.7, 182.5, 87.4, 86.5, 76.9, 76.8, 76.7, 73.1, 67.5, 67.4, 46.5, 45.6, 39.8, 38.4, 35.9, 35.6, 33.2, 33.1, 33.0, 32.9, 31.9, 29.3, 29.2, 26.3, 17.0. MS (ESI+) calculated for C_15_H_29_O_5_ [M + H]+ = 289.2, found = 289.0.

Typical methylenation procedure (using CO_2_)(4RS,5SR)-4-(hydroxymethyl)-3-methylene-5-2(octyldihydrofuran3H)-one, (±)-**UB006**

A 0.5 M LDA solution in THF (1.5 mL, 0.75 mmol) was slowly added to a stirred solution of (±)-*trans*-**7** (0.127 g, 0.6 mmol) in THF (5 mL) at −20 °C under N2. After 3 h at −20 °C, a CO_2_ (from solid CO_2_) flow was passed through the mixture for 30 min. The mixture was acidified with 6 N HCl to pH 2 and diluted with H_2_O (10 mL). The aqueous phase was extracted with CH_2_Cl_2_ (3 × 20 mL). The combined organic phases were dried (MgSO_4_), filtered, and concentrated under reduced pressure. The resulting brown residue was treated with a freshly prepared solution of acetic acid (1.03 mL), formalin (0.93 mL), sodium acetate (AcONa; 0.055 g), and N-methylaniline (0.52 mL), and the mixture was stirred for 2 h at room temperature. Then, 10 mL of saturated aqueous NaCl and concentrated HCl (1 mL) were added. After 5 min, the resulting residue was thoroughly extracted with CH_2_Cl_2_ (3 × 20 mL). The organic extracts were washed with brine, dried (MgSO_4_), filtered, and concentrated under reduced pressure. Purification of the residue through flash chromatography (CH_2_Cl_2_) yielded the desired product (72 mg, 0.315 mmol, 52%). Oil; Rf = 0.35 (CH_2_Cl_2_:MeOH 98:2); ^1^H NMR (400 MHz, CDCl_3_): δ 6.33 (d, J = 2.4, 1H), 5.71 (d, J = 2.4, 1H), 4.43–4.36 (m, 1H), 3.76 (dd, J = 6.4, 2.1, 2H), 2.91–2.83 (m, 1H), 1.87–1.59 (m, 2H), 1.55–1.17 (m, 12H), 0.88 (t, J = 6.9, 3H); ^13^C NMR (101 MHz, CDCl_3_): δ 170.5, 136.4, 123.6, 80.9, 64.1, 47.0, 36.2, 31.9, 29.5, 29.4, 29.3, 25.1, 22.8, 14.2. HRMS (ESI+) calculated for C_14_H_24_O_3_ [M + H]+ = 241.1798, found = 241.1793.

Typical methylenation procedure (using MMC)(4RS,5SR)-4-(1-hydroxyethyl)-3-methylene-5-octyldihydrofuran-2(3H)-one, (±)-**UB031**

A mixture of (±)-*trans*-**6** (0.135 g, 0.85 mmol) and 2 M dimethylmagnesium carbonate (MMC) in DMF (9.75 mL, 19.5 mmol) was stirred at 135 °C under N_2_ for 48 h. Subsequently, the solution was cooled to room temperature, and 6 N HCl (20 mL) and CH_2_Cl_2_ (20 mL) were carefully added. The layers were decanted, and the aqueous layer was extracted with further CH_2_Cl_2_ (3 × 20 mL). Then, the combined organic extracts were washed with aqueous saturated NaCl (50 mL), dried (MgSO_4_), filtered, and concentrated under reduced pressure. The resulting brown residue was treated with 3.5 mL of a freshly prepared solution of acetic acid (1.55 mL), formalin (1.48 mL), AcONa (0.05 g), and N-methylaniline (0.482 mL) and stirred at room temperature for 2 h. The reaction mixture was then treated with 10 mL of aqueous saturated NaCl and 1 mL of concentrated HCl. After 5 min, the resulting residue was thoroughly extracted with CH_2_Cl_2_ (3 × 20 mL). The organic extracts were washed with brine (30 mL), dried (MgSO_4_), filtered, and concentrated under reduced pressure. Purification of the residue through flash chromatography (CH_2_Cl_2_) yielded a sample of the desired product (16 mg, 0.067 mmol, 8%). Oil; Rf = 0.31 (CH_2_Cl_2_:MeOH 98:2); ^1^H NMR (400 MHz, CDCl_3_): δ 6.37 (d, J = 2.4, 1H), 5.73 (d, J = 2.4, 1H), 4.54 (m, 1H), 3.94 (m, 1H), 2.67 (m, 1H), 1.68–1.55 (m, 2H), 1.55–1.17 (m, 15H), 0.88 (t, J = 6.9, 3H); ^13^C NMR (101 MHz, CDCl_3_): δ 170.3, 136.4, 124.3, 79.3, 69.0, 51.4, 36.7, 31.8, 29.4, 29.3, 29.2, 25.0, 22.6, 20.1, 14.1. HRMS (ESI+) calculated for C_15_H_27_O_3_ [M + H]+ = 255.1955, found = 255.1958.

Preparation of (±)-**UB032**(4RS,5SR)-4-(1-hydroxypropyl)-3-methylene-5-octyldihydrofuran-2(3H)-one, (±)-**UB032**

A mixture of (±)-*trans*-**9** (0.230 g, 0.85 mmol) and 2 M dimethylmagnesium carbonate (MMC) in DMF (9.75 mL, 19.5 mmol) was stirred at 135 °C under N_2_ for 48 h. After cooling to room temperature, 6 N HCl (20 mL) and CH_2_Cl_2_ (20 mL) were added carefully to the solution. The layers were separated, and the aqueous layer was extracted with CH_2_Cl_2_ (3 × 20 mL). Then, the combined organic extracts were washed with NaCl (50 mL), dried (MgSO_4_), filtered, and concentrated under reduced pressure. The resulting brown residue was treated with 3.5 mL of a freshly prepared solution of acetic acid (1.55 mL), formalin (1.48 mL), AcONa (0.05 g), and N-methylaniline (0.482 mL) and stirred at room temperature for 2 h. The reaction mixture was then treated with 10 mL of aqueous saturated NaCl and 1 mL of concentrated HCl. After 5 min, the resulting residue was carefully extracted with CH_2_Cl_2_ (3 × 20 mL). The organic extracts were washed with brine (30 mL), dried (MgSO_4_), filtered, and concentrated under reduced pressure. Purification of the residue through flash chromatography (CH_2_Cl_2_) yielded a sample of the desired product as a mixture of stereoisomers (16 mg, 0.067 mmol, 8%). Oil; Rf = 0.31 (CH_2_Cl_2_:MeOH 98:2); ^1^H NMR (400 MHz, CDCl_3_): δ 6.35 (d, J = 2.4, 1H), 5.68 (d, J = 2.4, 1H), 4.58 (m, 1H), 3.66 (m, 1H), 2.71 (m, 1H), 1.64–1.49 (m, 2H), 1.50–1.41 (m, 2H), 1.55–1.17 (m, 12H), 1.00 (t, J = 7.0, 3H), 0.88 (t, J = 6.9, 3H); ^13^C NMR (101 MHz, CDCl_3_): δ 170.4, 137.0, 123.8, 79.1, 74.6, 50.0, 36.8, 31.7, 29.4, 29.3, 29.2, 26.8, 25.0, 22.7, 14.1, 10.1. HRMS (ESI+) calculated for C_16_H_29_O_3_ [M + H]+ = 269.2111, found = 269.2118.

Preparation of (±)-**UB033**(4RS,5SR)-4-(1-hydroxyheptyl)-3-methylene-5-octyldihydrofuran-2(3H)-one, (±)-**UB033**

A 2 M solution of THF LDA (0.73 mL, 1.47 mmol) was slowly added to a solution of (±)-*trans*-**11** (0.092 g, 0.29 mmol) in THF (5 mL) at −20 °C under N_2_, and the mixture was stirred for 3 h. A stream of CO_2_ (from solid CO_2_) was passed through the mixture for 30 min. The mixture was acidified with 6 N HCl to pH 2 and diluted with H_2_O (10 mL). The aqueous phase was extracted using CH_2_Cl_2_ (3 × 20 mL). The combined organic phases were dried (MgSO_4_), filtered, and concentrated under reduced pressure. The resulting brown residue was treated with 1.2 mL of a freshly prepared solution of acetic acid (0.5 mL), formalin (0.456 mL), AcONa (0.026 g), and N-methylaniline (0.255 mL) and stirred at room temperature for 2 h. The reaction mixture was then treated with 10 mL of aqueous saturated NaCl and 10 M HCl (1 mL). After 5 min, the resulting residue was thoroughly extracted with CH_2_Cl_2_ (3 × 20 mL). The organic extracts were washed with brine, dried (MgSO_4_), and concentrated under reduced pressure. Purification of the residue through flash chromatography (hexane/Et2O 6:4) yielded a sample of the desired product as a mixture of stereoisomers (5 mg, 0.015 mmol, 5%). Oil; Rf = 0.35 (hexane/Et2O 1:1); ^1^H NMR (400 MHz, CDCl_3_): δ 6.38 (d, J = 2.4, 1H), 5.69 (d, J = 2.4, 1H), 4.57 (m, 1H), 3.75 (m, 1H), 2.72 (m, 1H), 1.64–1.49 (m, 2H), 1.50–1.41 (m, 2H), 1.55–1.17 (m, 12H), 1.43 (s, 3H), 0.88 (t, J = 6.9, 3H) 0.92–0.80 (m, 8H); ^13^C NMR (101 MHz, CDCl_3_): δ 170.3, 137.0, 123.8, 79.0, 73.2, 50.3, 36.7, 33.7, 31.8, 31.7, 30.3, 29.7, 29.4, 29.3, 29.2, 29.1, 25.7, 24.9, 22.6, 14.1. HRMS (ESI+) calculated for C_20_H_37_O_3_ [M + H]+ = 325.2737, found = 325.2747.

Preparation of (±)-**UB034**(4RS,5SR)-4-(1-hydroxyallyl)-3-methylene-5-octyldihydrofuran-2(3H)-one, (±)-**UB034**

A 2 M solution of THF LDA (1.3 mL, 2.6 mmol) was slowly added to a solution of (±)-*trans*-**12** (0.136 g, 0.53 mmol) in THF (5 mL) at −20 °C under N_2_, and the mixture was stirred for 3 h. A stream of CO_2_ (from solid CO_2_) was passed through the mixture for 30 min. The mixture was acidified with 6 N HCl to pH 2 and diluted with H_2_O (10 mL). The aqueous phase was extracted with CH_2_Cl_2_ (3 × 20 mL). The combined organic phases were dried (MgSO_4_), filtered, and concentrated under reduced pressure. The resulting brown residue was treated with 2.2 mL of a freshly prepared solution of acetic acid (0.91 mL), formalin (0.82 mL), AcONa (0.048 g), and N-methylaniline (0.46 mL) and stirred at room temperature for 2 h. The reaction mixture was then treated with 10 mL of aqueous saturated NaCl and concentrated HCl (1 mL). After 5 min, the resulting residue was extracted with CH_2_Cl_2_ (3 × 20 mL). The organic extracts were washed with brine, dried (MgSO_4_), filtered, and concentrated under reduced pressure. Purification of the residue through flash chromatography (CH_2_Cl_2_) yielded the desired product as a mixture of stereoisomers (23 mg, 0.086 mmol, 16%). Oil; Rf = 0.15 (CH_2_Cl_2_); ^1^H NMR (400 MHz, CDCl_3_): δ 6.35 (t, J = 2.4, 1H), 5.87–5.70 (m, 2H), 5.35–5.32 (dt, J = 2.4, 1.3, 1H), 5.31–5.26 (m, 1H), 4.54–4.38 (m, 1H), 4.25–4.17 (m, 1H), 2.85–2.75 (m, 1H), 1.64–1.56 (m, 2H), 1.55–1.17 (m, 12H), 0.88 (t, J = 6.9, 3H); ^13^C NMR (101 MHz, CDCl_3_): δ 170.3, 170.1, 136.9, 136.6, 135.8, 135.4, 124.9, 124.6, 118.2, 118.1, 79.5, 79.2, 74.4, 74.3, 49.6, 49.5, 36.5, 36.3, 31.8, 29.3, 29.2, 29.1, 24.9, 24.8, 22.6, 14.1. HRMS (ESI+) calculated for C_16_H_27_O_3_ [M + H]+ = 267.1955, found = 267.1953.

Preparation of (±)-**UB035**(4RS,5SR)-4-((RS)-1,2-dihydroxyethyl)-3-methylene-5-octyldihydrofuran-2(3H)-one, (±)-**UB035**

A 2 M solution of THF LDA (1.82 mL, 3.64 mmol) was slowly added to a stirred solution of (±)-*trans*-**13** (0.189 g, 0.73 mmol) in THF (5 mL) under N_2_ at −20 °C. After 3 h, a stream of CO_2_ (from solid CO_2_) was passed through the mixture for 30 min. The mixture was acidified with 6 N HCl to pH 2 and diluted with H_2_O (10 mL). The aqueous phase was extracted with CH_2_Cl_2_ (3 × 20 mL). The combined organic phases were dried (MgSO_4_), filtered, and concentrated under reduced pressure. The resulting brown residue was treated with 3 mL of a freshly prepared solution of acetic acid (1.25 mL), formalin (1.13 mL), AcONa (0.065 g), and N-methylaniline (0.632 mL) and stirred at room temperature for 2 h. The reaction mixture was then treated with 10 mL of saturated NaCl and concentrated HCl (1 mL). After 5 min, the resulting residue was thoroughly extracted with CH_2_Cl_2_ (3 × 20 mL). The organic extracts were washed with brine, dried (MgSO_4_), and concentrated under reduced pressure. Purification of the residue through flash chromatography (CH_2_Cl_2_/MeOH 99:1) yielded the desired product as a mixture of stereoisomers (43 mg, 0.16 mmol, 22%). Oil; Rf = 0.25 (CH_2_Cl_2_:MeOH 95:5); ^1^H NMR (400 MHz, CDCl_3_): δ 6.19 (d, J = 2.4, 1H), 5.70 (d, J = 2.4, 1H), 4.58 (m, 1H), 3.62 (m, 1H), 3.47 (m, 2H), 2.87 (m, 1H), 1.52 (m, 2H), 1.55–1.17 (m, 12H), 0.88 (t, J = 6.9, 3H); ^13^C NMR (101 MHz, CD_3_OD): δ 171.2, 137.6, 122.9, 79.2, 73.2, 46.8, 36.1, 31.6, 29.1, 29.0, 28.9, 26.8, 24.6, 22.3, 13.0. HRMS (ESI+) calculated for C_15_H_27_O_4_ [M + H]+ = 271.1904, found = 271.1912.

Preparation of (±)-**UB036**(4RS,5SR)-3-methylene-5-octyl-4-(1,2,3-trihydroxypropyl)dihydrofuran-2(3H)-one, (±)-**UB036**

A solution of (±)-*trans*-**14** (0.134 g, 0.46 mmol) and 2 M solution of dimethylmagnesium carbonate (MMC) in DMF (8.13 mL, 16.2 mmol) was stirred at 135 °C under N_2_ for 48 h. Subsequently, the solution was cooled to room temperature and 6 N HCl (20 mL) and CH_2_Cl_2_ (20 mL) were added carefully. The layers were decanted, and the aqueous layer was extracted with additional CH_2_Cl_2_ (3 × 20 mL). Then, the combined organic extracts were washed with aqueous saturated NaCl (50 mL), dried (MgSO_4_), filtered, and concentrated under reduced pressure. The resulting brown residue was treated with 3.5 mL of a freshly prepared solution of acetic acid (1.29 mL), formalin (1.23 mL), AcONa (0.042 g), and N-methylaniline (0.402 mL) and stirred at room temperature for 2 h. The reaction mixture was then treated with 10 mL of saturated aqueous NaCl and 1 mL of concentrated HCl. After 5 min, the resulting residue was thoroughly extracted with CH_2_Cl_2_ (3 × 20 mL). The organic extracts were washed with brine (30 mL), dried (MgSO_4_), filtered, and concentrated under reduced pressure. Purification of the residue through flash chromatography (CH_2_Cl_2_/MeOH 94:6) produced the desired product as a mixture of stereoisomers (70 mg, 0.24 mmol, 53%). Oil; Rf = 0.2 (CH_2_Cl_2_/MeOH 94:6); ^1^H NMR (400 MHz, CDCl_3_): δ 6.15 (s, 1H), 5.57 (s, 1H), 4.55 (m, 1H), 3.67–3.27 (m, 4H), 2.97 (m, 1H), 1.55–1.17 (m, 12H), 0.88 (t, J = 6.9, 3H) ^13^C NMR (101 MHz, CDCl_3_): δ 172.5, 172.4, 138.9, 135.7, 125.3, 123.3, 82.6, 79.6, 74.7, 73.5, 72.0, 71.4, 64.4, 47.0, 46.9, 37.0, 36.6, 33.2, 33.1, 32.9, 29.3, 25.4, 25.3, 14.3. HRMS (ESI+) calculated for C_16_H_29_O_5_ [M + H]+ = 301.2010, found = 301.2020.

### 3.2. Cell Culture and Viability Assays

OVCAR-3 (human ovary carcinoma, ATC HTB-161, passage number 20) cells were cultured at 37 °C in a humidified atmosphere of 5% CO_2_ in Dulbecco’s Modified Eagle Medium containing 4.5 g/L of glucose and supplemented with glutamine, 10% heat-inactivated fetal bovine serum, 100 units/mL of penicillin, and 100 mg/mL of streptomycin. The culture was passaged twice a week through gentle trypsinization. Cells were seeded in 10 cm culture dishes and grown to confluence. An MTT cytotoxicity assay was performed to evaluate the cytotoxic effect of the compounds. A total of 2 × 103 cells/well were plated in 96-well plates in 100 mL of culture medium. After 24 h, the medium was removed, and cells were incubated for 72 h in fresh medium with different concentrations (0.5–20 µM) of the different compounds. DMSO was used as a blank at a final concentration of 0.5%. The cells were then incubated for 3 h with 100 mL of fresh medium and 20 mL of MTT dissolved in phosphate-buffered saline to a final concentration of 5 mg/mL. Following treatment, the supernatants were carefully removed, and the MTT–formazan crystals were solubilized by adding 100 mL/well of DMSO. The absorbance was measured at 570 nm.

### 3.3. Turbidimetry Assay

The compound (±)-**UB006** and its derivatives were dissolved in 300 µL of milliQ-water at 1% DMSO in a 96-well plate to concentrations ranging from 25 to 2000 µM. The compounds were incubated for 2 h at 37 °C in a Thermoshaker (Biosan^®^, Riga, Latvia). Solubility was assessed through turbidimetry at 600 nm using Varioskan Lux^®^ (Thermo Scientific^®^, Waltham, MA, USA). The approximate solubility was calculated by finding the intersection of the two regression lines formed before and after the curve tendency change due to precipitation of the compounds.

### 3.4. FAS Activity

For in vitro experiments, FAS was purified from rat liver following the protocol previously described by Linn [19]. FAS activity was measured using a spectrophotometric method, as previously described [20]. Cytosolic hepatic extracts obtained from rat (210 µg) were preincubated at 30 °C for 30 min in a 96-well plate with increasing concentrations of the following, (±)-**UB006**, (±)-**UB035**, and (±)-**UB036** dissolved in DMSO (100–5000 mM), using DMSO as a blank. The remaining derivatives were not tested based on cytotoxicity results obtained in the MTT assay. NADPH (121 µM) and acetyl-CoA (61 µM) in potassium phosphate buffer (pH 7.2) were added to preincubated enzyme and equilibrated at 37 °C for 3 min. The reaction was initiated through the addition of malonyl-CoA (55 µM). The total reaction volume was 330 µL. Oxidation of NADPH was measured at 340 nm at 37 °C for 10 min.

### 3.5. Statistical Analysis

Data are expressed as the mean ± SEM. Data processing was assessed using linear regression and non-linear variable slope regression. Statistical analysis was assessed using one-way ANOVA. GraphPad Prism 10.0 was used for data representation.

## 4. Conclusions

We aimed to identify more potent and soluble anti-cancer compounds by synthesizing several derivatives of (±)-**UB006** via elongation of the carbon chain of the primary alcohol and/or the addition of hydroxyl groups. We have accomplished the synthesis of a family of highly soluble (±)-**UB006** derivatives that show promising activity as anti-cancer agents. We conclude that (±)-**UB035** showed higher cytotoxic activity and solubility than (±)-**UB006**. Remarkably, while incorporation of an ethyl secondary hydroxyl group to (±)-**UB006** did not display a higher inhibitory effect on FAS activity, it significantly increased the in vitro antitumor effects against the ovarian cancer cell line, OVCAR3. This addition also dramatically improved solubility, which is a major shortcoming of **C75** and previously synthesized derivatives, and it also created a new stereocenter. It has been described that FAS inhibitors trigger cell death through apoptotic mechanisms [21,22]. In fact, we have previously shown that the mechanisms through which (−)-**UB006** exerts its cytotoxic effect on the OVCAR-3 cell line include a decrease in mitochondrial respiratory capacity and ATP production, an upregulation of the energy stress factor DDIT4/REDD1, mTOR activity inhibition, and caspase-3 activation, leading to apoptosis [18]. DDIT4/REDD1 is known to inhibit the mTOR pathway and cell proliferation [23]. The cytotoxicity effect of the new (±)-**UB006** derivatives described here might involve similar mechanisms that finally trigger cancer cell apoptosis. Both (±)-**UB035** and (±)-**UB036** derivatives showed higher solubility compared to (±)-**UB006** (Table 2). Although this increase in solubility did not change FAS’s inhibitory effects compared to (±)-**UB006**, it might have contributed to their enhanced cytotoxicity in the ovarian cancer cells tested. Future research should include the study of (±)-**UB035** stereoisomers in vivo. However, we believe that their effects will likely correlate with those of (±)-**UB006**. Furthermore, the stereoisomers derived from this new chiral center may show different cytotoxicity, and further investigation into these compounds is required for the treatment of cancer.

## Data Availability

Data generated during the study are available from the corresponding author upon reasonable request.

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
