# Peer review of "New UB006 Derivatives With Higher Solubility and Cytotoxic Activity in Ovarian Cancer Cells"

_pharmaceuticals, 2025, doi:10.3390/ph18020194_

Round 1

Reviewer 1 Report

Comments and Suggestions for Authors

In this manuscript, the authors synthesized several (±)-UB006 derivatives by elongating the carbon chain of the primary alcohol and/or adding hydroxyl groups. (±)-UB035 was found to be the most promising compound with a 2.5-fold increase in cytotoxicity in OVCAR-3 cell line and a >4-fold increase in solubility (>2 mM) compared with (±)-UB006. One limitation of this work is that only one cell line was tested for the synthesized compounds. Second, the authors conclude that the enhanced cytotoxic activity observed for (±)-UB035 compared with (±)-UB006 was not attributable to differences in FAS inhibition, but no further in vitro study was performed to clarify the mechanism of action for these compounds. Overall, the manuscript is well written, I recommend to consider this work for publishing in Pharmaceuticals after addressing the following points.

1. Figure 1. The stereocenters in the structures should not draw like this, please follow the authors previous publication (Eur J Med Chem, 2017, 131, 207-221)

2. I suggest the authors give the chemical structure of (±)-C75 in Figure 1 since C75 was mentioned so many times in the manuscript.

3. Section 2.3. The authors mentioned they analyzed the effect of racemic mixtures of (±)-C75, (±)-UB006, and its derivatives (±)-UB035, and (±)-UB036 on FAS activity, but the FAS inhibition data of (±)-C75 was not shown in Figure 6.

4. Section 3.1, The authors should give the mass data for the intermediates and target compounds.

5. Figure 2-4, The figures should not have the same titles.

6. I suggest the authors use Scheme 1-3 instead of Figure 2-4 for the reaction schemes.

7. Figure 3. The reaction procedure letters shown in the figure (l, m, n, etc.) are inconsistent with the footnotes (a, b, c, etc.)  

8. Figure 4. The authors should give the reaction yields in the footnote text.

9. There are numerous grammatical errors and typos, like IC50, CH2 in Table 1, >2mM in Table 2, a careful reading of the manuscript should be performed.

Author Response

Please, see attached pdf.

Reviewer 2 Report

Comments and Suggestions for Authors

In this MS, different organic derivatives of UB006 compounds were synthesized and tested for their efficacy on ovarian cancer cell lines  (OVCAR 3) through cytotoxicity test. It is an important piece of work in the development of cancer therapy. The authors should clarify the following comments and revise the MS accordingly

1. In this MS, authors used just only one cancer cell line (OVCAR 3), so the title should  be more precise with exact type of cancer what it was done exactly. The usage of the term "Anti-cancer compounds' which deals in a wide range of cancers. In this investigation, compounds were tested only on ovarian cancer cells. Re-write the title accordingly.

2. In the abstract background, it was mentioned "We previously showed", this sentence should be used in the introduction review along with the references.

3. In many places, starting word in the sentences with abbreviation were noticed frequently (ex. abstracts line 13, sentence start with (+) UB006; Line 49, 56 ---- sentences start with C75, in the methodology, most of the paragraph sentences starts with similar way. This kind of grammatical errors should be corrected completely during revision. 

4. Similarity index percent is very high in the methodology section. Through re-writing, It should be minimized.

5. Though the methods mentioned elaborately, results and discussions were lack of explanation, interpretation, comparative statements. Just about 4 citation of references used in this section. Data obtained (figures, tables etc, ) In this investigation should be explained and supported by many other reports.

6. What is the passage number for the Cell line used in this investigation? Because, the Passage number is important for the cell lines to restore their normal functions. 

7. In this investigation, the control was not maintained In the cytotoxicity assay? Toxicity was tested only on cancer cell lines and not on normal cells. Toxicity of the compounds should be also tested on normal/non-cancer cells. Because the functions of cancer and non-cancer/normal cells are not same. 

9. In figure No.5, there are 2 graphical pictures were given. In the caption it was defined distinctly. Differences on both graphs should be cited in the caption.

10. Discussions should added more information based on the results. Earlier works/reports should be added for comparison and interpretation.

Author Response

Please, see attached pdf.

Round 2

Reviewer 1 Report

Comments and Suggestions for Authors

The authors addressed the points reported in my review, the manuscript can be published.

Reviewer 2 Report

Comments and Suggestions for Authors

accept in present form